# Macroscopic Friction Studies of Alkylglucopyranosides as Additives for Water-Based Lubricants

**Wei Chen [1], Tobias Amann [2] 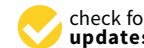, Andreas Kailer [2] and Jürgen Rühe [1,*]**

[1] IMTEK—Department of Microsystems Engineering, University of Freiburg, Georges Köhler Allee 103, 79110 Freiburg, Germany; wei.chen@imtek.uni-freiburg.de

[2] Fraunhofer Institute for Mechanics of Materials IWM, Wöhlerstraße 11, 79108 Freiburg, Germany; tobias.amann@iwm.fraunhofer.de (T.A.); andreas.kailer@iwm.fraunhofer.de (A.K.)

[*] Correspondence: ruehe@imtek.uni-freiburg.de

**Abstract:** Water-based lubricants might become an interesting alternative to conventional oil-based lubricants and help to reduce wear as well as improve the energy efficiency of transport processes. Since pure water is generally a rather poor lubricant due to its low viscosity and corrosiveness, it must be tribologically optimized with suitable additives. Here, we study the friction behavior of alkyl glucopyranosides (AGPs) with varying lengths of the alkyl chain. Sliding experiments show that a significant reduction in the coefficient of friction compared to that of pure water is observed. The extent of friction reduction depends strongly on the concentration and on the shearing conditions. It is assumed that the low coefficients of friction are due to the ability of AGPs to form liquid crystalline phases with an ordered structure in the friction gap. Furthermore, the interaction of the AGPs with the surface forms a wear protection layer (boundary lubrication). The friction properties of the water-based system are compared to those of a conventional, mineral oil-based lubricant.

**Keywords:** water-based lubricant; liquid crystal; surfactant; wear; thin film lubrication

## 1. Introduction

Tribological contacts account for about 23% (119 EJ) of total global energy consumption. Twenty percent of this is used to overcome friction and 3% is used to repair wear failures [1–5]. With rising awareness of sustainability, current tribological research therefore focuses on the development of sustainable and energy-efficient systems ("green tribology") [6,7]. New surface and material technologies, the substitution of mineral oil-based lubricants with biodegradable ones or the mimicking of natural systems allow new tribological systems to be developed [6,8].

In this context, water-based lubricants could be an option to replace mineral oils in certain applications, due to their high fluidity and high thermal conductivity [9]. On the other hand, the corrosiveness and low viscosity of water pose serious challenges in this context. Therefore, additives are currently being sought which improve the properties of water as a lubricant. Some fundamental studies of water-based lubricants at the nanoscale have been investigated to provide a deep insight into the lubrication mechanism. An ultralow COF can be achieved through the use of surfactants [10,11] as additives or by employing surfaces decorated with polymer brushes [12,13]. The extremely low friction was ascribed to efficient lubrication between hydrated layers [14]. Others studied nanostructured materials, such as metals, metal oxides and graphene, as additives to water in this context [15–20]. However, in these cases the application of the lubricant was limited by their poor dispersibility. Efforts were made to reduce their high surface activity and avoid aggregation [20]. Fluorinated graphene was functionalized by urea to improve surface wettability and to ensure a homogenous dispersion

of the additive in water. The layer structure of graphene and the weak van der Waals force between the adjacent layers allow for a low shear stress and thus low friction [19]. Zhang et al. have modified Cu nanoparticle surface with xanthate to promote the dispersibility and thus lower the friction. It is assumed that Cu nanoparticles form a wear-protecting boundary film through a tribochemical reaction with the steel surface, which leads to an improvement in the frictional properties [17]. In addition, some polymers (e.g., polyalkylene glycol [21]), ionic liquids (e.g., ibuprofen-based ionic liquids [22]) and surfactants (e.g., ammonium bromide cationic surfactant [23]) were identified as effective additives to form a protective film, offering better sliding properties and wear protection due to their special physicochemical properties. In case of an ionic liquid, for instance, imidazolium/phosphonium-based ionic liquids [24–26] and NaCl solutions [27] the surface adsorption can also be controlled by application of a galvanic potential.

Investigations into complex fluids with anisotropic properties, e.g., mesogenic fluids, ionic liquids and ionic liquid crystals, have shown their potential for friction reduction [28,29]. Liquid crystals (LCs) are a further substance class of additives which have been shown to improve friction and optimize water as a lubricant. For example, LCs are contained in the human synovial fluid and are essential for the good friction behavior of joints. The adsorption and alignment of cholesterol on the cartilage surfaces reduces friction during a sliding motion [30].

Behavior where the viscosity in sliding direction is reduced while the viscosity perpendicular to the sliding direction, i.e., in the direction of the load, is rather high, is frequently called thin-film lubrication. LCs have already been intensively investigated as pure lubricants [31] and as additives in mineral oils [32,33] and water [34,35]. A new field of research is currently opening up in the field of ionic liquid crystals (ILCs) [29], also in combination with water [36,37]. It is assumed that the resulting ultralow friction and wear are ascribed to their specific molecular interactions and orientation in the shear flow [38]. The liquid crystalline mesophase is characterized by the anisotropy of physical properties, especially, in this case, the viscosity. In the substance class of lyotropic liquid crystals, the formation of the liquid crystalline phase depends on concentration, temperature and pressure. Due to its anisotropic properties, combined with the ability to form stable surface layers, these lyotropic LCs are promising compounds for tribological applications [39,40]. Ma et al. and Fuller et al. have formulated a mixture with liquid crystal structure, in order to test their tribological performance. The lamellar liquid crystals show lower coefficients of friction (COF) with a higher load-carrying capacity compared to the commercial lubricants [41,42]. Boschkova et al. proposed that good lubrication is observed when a lamellar liquid crystalline phase is dispersed in water. It adsorbs on the steel surface and forms a lubricating tribofilm. When this film is ruptured at high pressure, the low viscosity of the dispersed lamellar phase allows for a fast relaxation and the formation of a new tribofilm [34]. Combined lamellar crystalline phase and nanostructure materials have also been studied by Yang et al. with respect to their anti-friction properties. Such a system can be viewed as stiff, thin boards with small balls in between them The rigid balls allow that the board-like lamellar phase can slide against each other more easily, not unlike a ball-bearing [43–45].

However, these additives might only have limited potential in large scale applications since they are even less environmentally friendly than mineral oils and might pose a significant threat to the environment and ecology. "Green" water-based lubricants have hence won wide interest. Ji et al. have investigated the tribological properties of a water-based system derived from renewable lignocellulosic biomass [46]. Sulek et al. have developed a biocompatible aqueous system of passionfruit oil ethoxylate. The low movement resistance and high load-bearing capacity of such systems are the result of the adsorption of a protective layer with ordered structure, which is concluded as an adsorptive-structural mechanism [47]. In addition, they have used biodegradable multi-component solutions based on alkyl polyglycosides (APGs) as a component to reduce friction and wear [48,49]. APGs with a longer alkyl chain length (C12-14) have a high surface affinity and reduce motion resistance and wear more effectively, while APGs C8–C10 have better anti-seizure properties [49].

However, APGs are generally composed of compounds with varying substituent chain lengths, and, additionally, vary in their molecular weight, and therefore the tribological mechanisms cannot be easily understood. In this work, structurally well-defined alkyl glucopyranosides were used for rheological and tribological analyses. In a previous study, low COF in a binary octyl-ß-D-glucopyranoside (C8)/$H_2O$ system with a concentration of 40% was already demonstrated [50]. It was assumed that the interaction between the surface and the shear-induced molecular orientation is responsible for the improvement in tribological properties. The aim of this work is to investigate the substance class of glucopyranosides in more detail. Therefore, different concentrations of C8 in water were investigated with respect to their critical micelle concentration (CMC). In addition, further glucopyranosides with shorter (C6) and longer (C10) alkyl chain length were investigated to analyze the influence on friction. The results are discussed in comparison to an aqueous mixed solution of alkylated oligomeric sugar (C12–C14) and reference oil.

## 2. Materials and Methods

### 2.1. Materials

Aqueous systems based on Octyl ß-D-glucopyranoside (C8), hexyl ß-D-glucopyranoside (C6) and decyl ß-D-glucopyranoside (C10) were used as model substances to study the frictional behavior and the influence of alkyl chain length (Sigma-Aldrich GmbH, Darmstadt, Germany). In addition, Glucopon 600 CSUP (Sigma-Aldrich GmbH, Darmstadt, Germany), an approximately 50% aqueous mixed solution of alkylated oligomeric sugar (C12–C14) was used. As a comparison for friction coefficient behavior, Optigear 32 (Castrol, Freiburg, Germany) was tested, which is a high-performance gear oil for long-term lubrication. The aqueous solutions with different concentrations in weight percent were prepared by solving in millipore water (0.05 µS/cm, Simplicity®, Germany). 100Cr6 plate (Optimol Instruments, Munich, Germany) served as a friction surface for the tribological measurements.

### 2.2. Methods

Static contact angle measurements (OCA20 Setup by Dataphysics GmbH, Filderstadt, Germany) on a 100Cr6 steel surface were carried out to determine the CMC of C8 aqueous solution. The dosing volume is 5 µL with a dosing rate of 2 µL/s. A mean value was calculated from at least five positions.

The viscosity of all samples was determined by a rotational rheometer (Physica MCR 501, Anton Paar, Ostfildern, Germany) with cone-plate geometry (CP: 50-2/TG, diameter: 49.915 mm, angle: 2.001°) at an increasing shear rate of 0.1 $s^{-1}$ to 100 $s^{-1}$ at 20 °C.

The ability of alkyl glucopyranoside with different alkyl chain lengths to form ordered liquid crystalline structures as a function of concentration was investigated. The phase behavior at room temperature was characterized by light microscopy (Scope A1, Zeiss, Oberkochen, Germany) under crossed polarizers with an LED light source.

Tribological experiments were performed with ring (100Cr6, outer diameter: 20 mm, inner diameter: 17 mm, 60 HRD, *Ra*: 0.22 µm, *Rz*: 1.87 µm)-on-plate (100Cr6, diameter: 24 mm, height: 6.9 mm, 62 HRD, *Ra*: 0.047 µm, *Rz*: 0.63 µm)-sliding geometry (SRV-4, Optimol Instruments, Munich, Germany). To investigate the effect of concentration on tribological performance, C8 was prepared with concentrations of 0.44%, 0.73%, 1.02%, 10%, 20%, 30% and 40%. The tribological tests were performed at 50 N (0.6 MPa) and 50 Hz (0.1 m/s) for 1 h with stroke 1 mm at room temperature. Frequency ramp measurements were also carried out at oscillating speeds from 10 to 75 Hz (0.02–0.15 m/s) under modified load at room temperature and the results of different materials (alkyl glucopyranoside with different alkyl chain length, Glucopon 600 CSUP and Optigear 32) were compared. For each frequency level the test duration was 5 min. For each measurement, 1 mL sample was dropped between ring and plate, as shown in Figure 1.

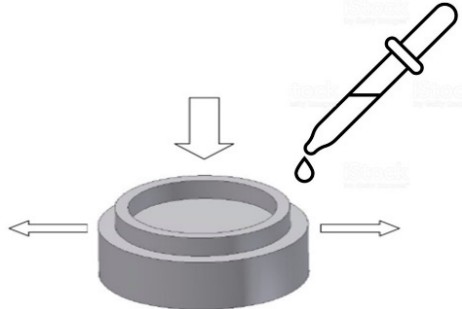

**Figure 1.** Schematic set-up of ring-on.plate geometry.

The surface after the friction test was characterized by 3D laser microscopy (Keyence VK9700, Japan).

## 3. Results

### 3.1. Physicochemical Properties

The wettability of C8 aqueous solution on steel surfaces and its critical micelle concentration (CMC) was determined using static contact angle measurements (Figure 2). The CMC is defined as the surfactant concentration above which micelles form. At this concentration, a stable surface energy with an almost constant static contact angle is reached. As shown in Figure 2, the contact angle decreases from 38° at 20 mM concentration in a linear fashion to a nearly constant value of 21° as the concentration reaches 25 mM.

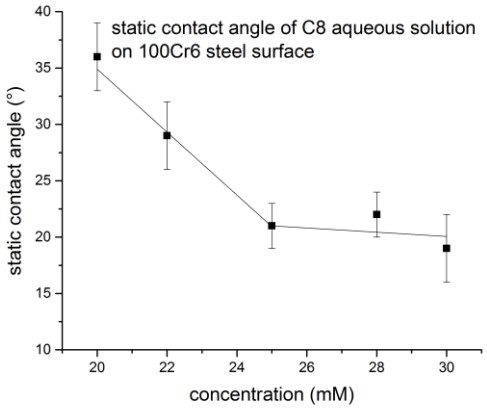

**Figure 2.** Static contact angle of C8 aqueous solutions on 100Cr6 steel surface; 5 µL, *T* = 25 °C.

The viscosity of C6, C8, C10 and Glucopon 600 CSUP aqueous solutions were recorded and compared in Figure 3. All solutions had the same concentration of 40%. It is seen that the viscosity increases with increasing alkyl chain length. For instance, the viscosity of C10 aqueous solution is 90 times and 13 times higher than those of C6 and C8 solutions, respectively, at the chosen shear rate of 100 s$^{-1}$. The Glucopon solution possesses the highest viscosity since it contains various APG oligomers and other additives. The high-performance oil, Optigear 32, with approximately 75 mPa·s, has a comparable viscosity, at 100 s$^{-1}$, as the 40% C8 aqueous solution with 29 mPa·s.

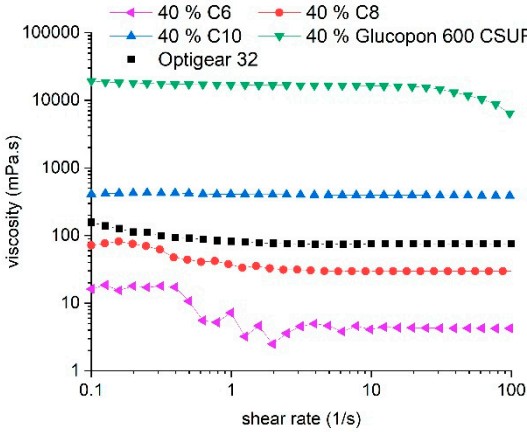

**Figure 3.** Dynamic viscosity of C6, C8, C10, Glucopon 600 CSUP aqueous solution with 40% and Optigear 32, at 20 °C.

The ability of alkyl glucopyranoside to form a liquid crystal and its dependency on concentration was studied with polarized light microscopy at room temperature. The microscopic images are shown in Figure 4.

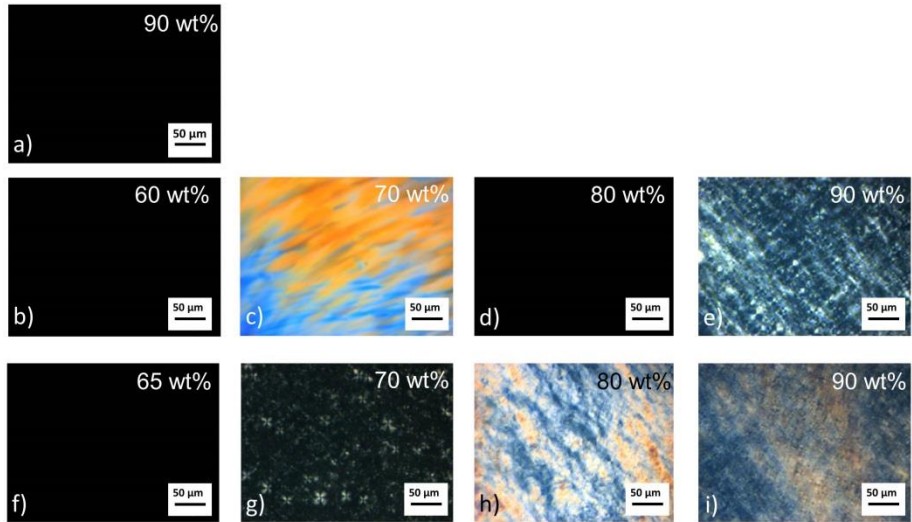

**Figure 4.** Microscopic images of C6, C8 and C10 aqueous solution with different concentrations under a polarized light microscope at room temperature: (**a**) 90% C6 aqueous solution; (**b**–**e**) 60–90% C8 aqueous solution [50]; (**f**–**i**) 65–90% C10 aqueous solution.

The dark color of the microscopy image taken with 90% C6 aqueous solution in Figure 4a implies that the C6 solution is optically isotropic over the whole concentration range. In aqueous solution C8 molecules form micelles, which have an ordered hexagonal phase with a typical fan-like texture, until the concentration reaches ~70%, as seen in Figure 4c. Thereafter, a further increase in concentration leads to the formation of a cubic phase, which is optically isotropic (Figure 4d). Finally, at a concentration of 90%, we observe the formation of a lamellar phase, which features a gel-like oily streak (Figure 4e). In comparison to the shorter chain analogs, the C10 molecules aggregate and begin to form a lamellar phase over a wide concentration range, about 70–100%. As the concentration increases, the liquid crystal phase gets more viscous, as seen in Figure 4g–i. In summary, the optical results of C6, C8 and C10 under polarized light microscopy suggest that C10 form lyotropic liquid crystalline phases over a much wider concentration range compared to the other compounds.

### 3.2. Friction Coefficient Behavior of Lubricants Based on Glucopyranoside

Figure 5a shows that a remarkable decrease in the COF of water, which is around COF = 0.4 at the chosen load and sliding speed, is observed after adding a little C8. Already, after the addition of very low amounts of C8 (concentration ~0.5%), the friction drops to about half and decreases further until at the CMC (25 mM (0.73%)), where a value of COF = 0.15 is reached. Addition of 1% of the lubricant already reduces the friction by almost 70%. Such a low concentration of the alkyl sugar is sufficient to form an adsorbed layer on both sliding surfaces. The frictional properties remain more or less unchanged with increasing concentration, up to a concentration of more than 10%. When an even higher concentration is chosen, a further decrease in the COF, to about 0.04, can be observed (Figure 5b).

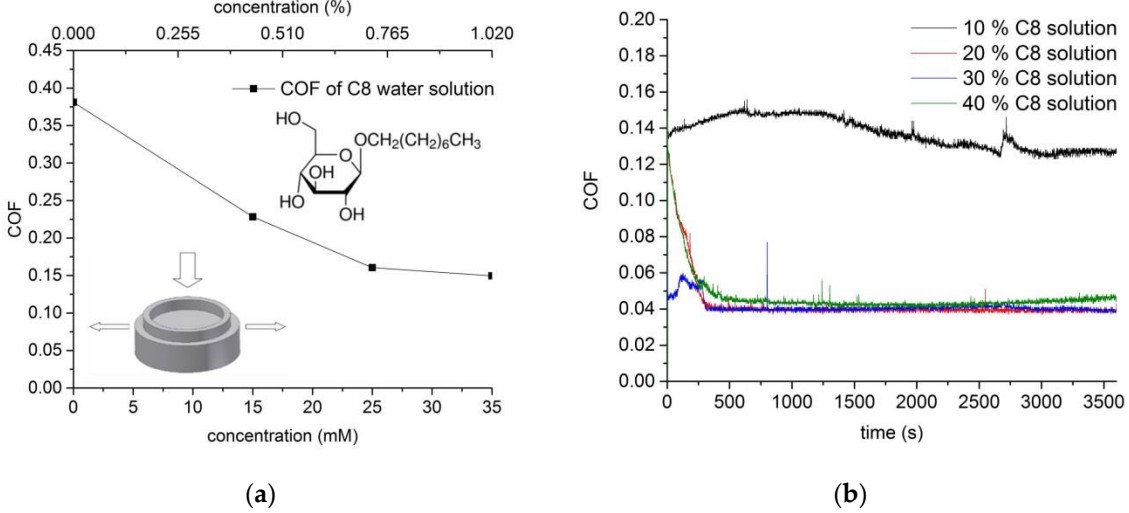

(**a**)    (**b**)

**Figure 5.** (**a**) Friction coefficient behavior of C8 tested with ring plate geometry as function of concentration (0–1.02%); (**b**) Coefficients of friction (COF) of C8 aqueous solution with a concentration of 10–40% over time (test conditions: friction pair 100Cr6-100Cr6, 50 N (0.6 MPa), 50 Hz (0.1 m/s), 1 mm stroke, 1 h).

The influence of the alkyl chain length on the friction coefficient behavior was investigated using 40% C6, C8 and C10 aqueous solutions at 100 N (1.2 MPa) as the sliding speed was increased from 0.02 m/s to 0.15 m/s. The measurement results are presented in Figure 6.

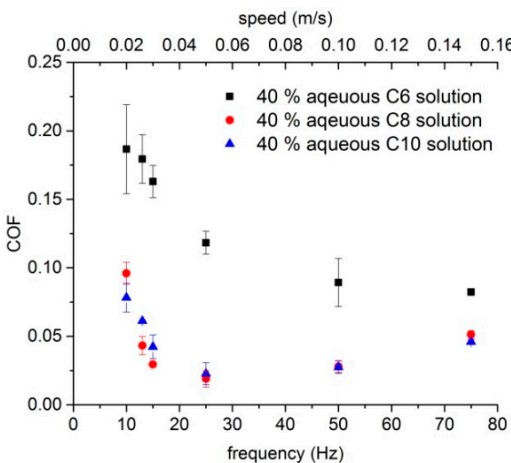

**Figure 6.** Friction properties of 40% C6, C8 and C10 aqueous solutions measured with ring-plate geometry at varying sliding speeds (0.02–0.15 m/s) (test conditions: friction pair 100Cr6–100Cr6, 100 N (1.2 MPa), 1 mm stroke).

40% C8 and C10 aqueous solution have a similar COF as a function of sliding speed. From 10 to 25 Hz, the COF decreases steadily and a minimum of approximately 0.02 is achieved. As frequency of the shearing process is increased, the COF rises to about 0.05. In comparison, the COF of 40% C6 aqueous solution is higher over the whole range of sliding velocities. It decreases only from 0.18 to 0.08 as the sliding speed increases from 0.02 to 0.15 m/s.

To investigate the wear behavior of the glucopyranoside/water lubricated surfaces the steel (100Cr6—100Cr6) surfaces underwent a 1 h friction test (100 N (1.2 MPa) and 25 Hz (0.05 m/s)) and were subsequently characterized by microscopy. The obtained images are presented in Figure 7.

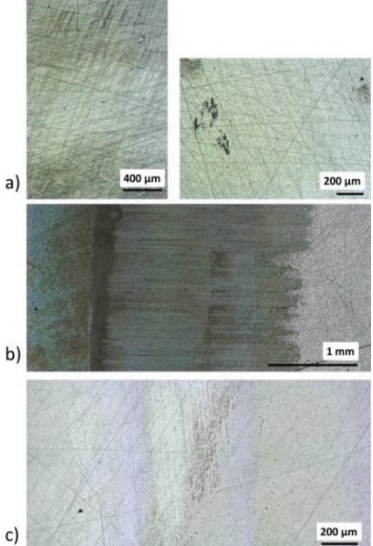

**Figure 7.** Characterization of steel surfaces after 1 h friction investigation with (**a**) 40% C6 solution, (**b**) 40% C8 solution [50] and (**c**) 40% C10 solution under oscillating motion with friction pair 100Cr6–100Cr6 at 100 N (1.2 MPa) and 25 Hz (0.05 m/s), stroke 1 mm.

For both the surfaces lubricated with 40% C6 and C10 aqueous solutions, a wear track was clearly visible, accompanied by slight tribocorrosion on the surface (Figure 7a,c). This indicates that, due to wear, the surface is exposed to the aqueous environment, and accordingly the corrosion process is stimulated. In contrast, in the case of the C8 solution, Figure 7b, no such wear tracks are visible and only a C8 film is visible on the surface, which implies a strong interaction between the C8 molecules and the steel surface.

In addition, the C8 sample was compared with the Glucopon 600 CSUP sample with the same concentration of 40%. This is a mixture of alkyl polyglycosides with varying chain lengths. The measurement conditions were the same as before and the results are shown in Figure 8.

In the 40% Glucopon solution, a strong reduction in the COF to values approximately 0.06–0.07 (at >0.03 m/s) is noticed compared to pure water. However, the friction is still a little higher than that of the C8 solution under the same measurement conditions. For example, at 25 Hz the COF of C8 solution reaches 0.02, while in the Glucopon solution it is about 0.07.

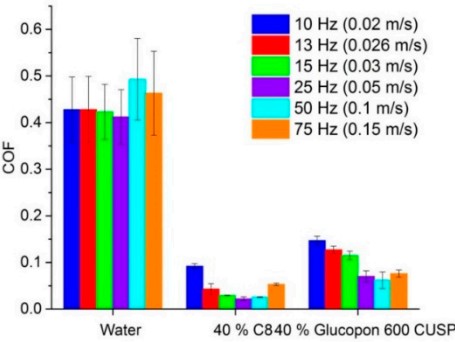

**Figure 8.** Comparison of tribological performance between water, 40% C8 aqueous solution and 40% Glucopon 600 CSUP at 100 N (1.2 MPa) and different oscillating frequencies 10–75 Hz (0.02–0.15 m/s) with friction pair 100Cr6–100Cr6, stroke 1 mm.

*3.3. Comparison with a Standard Oil*

The friction coefficient behavior of the model substance C8 was compared with the standard oil Optigear 32, which has a similar viscosity as a 40% C8 aqueous solution. The measurements were performed at the load 50–150 N (0.6–1.8 MPa) with sliding frequencies of 10–75 Hz (0.02–0.15 m/s). The friction results are presented in Figure 9.

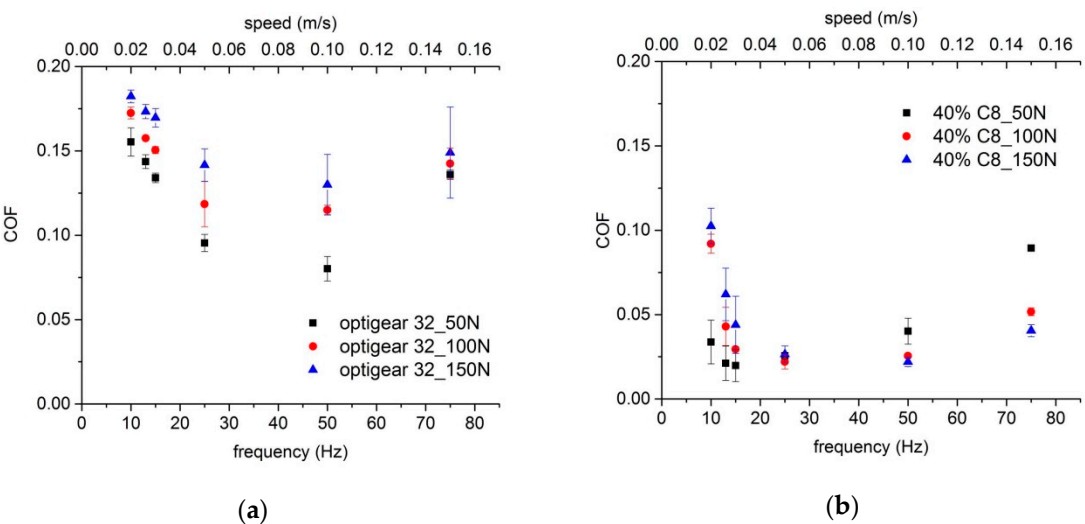

**Figure 9.** Tribological performance of (**a**) commercial gear oil Optigear 32 and (**b**) 40% C8 aqueous solution [50], at load 50–150 N (0.6–1.8 MPa) and different oscillating frequencies in the range 10–75 Hz (0.02–0.15 m/s) with friction pair 100Cr6–100Cr6, 1 mm stroke.

Figure 9a shows that the minimum COF of the commercially available lubricant Optigear 32 under the applied conditions is around 0.08. The COF decreases initially with increasing sliding speed, and then increases again at 50 Hz. With higher loads, from 50 N (0.6 MPa) to 150 N (1.8 MPa), the COF rises, e.g., from 0.08 to 0.13 at 50 Hz. Under same measurement conditions, the friction curve of the 40% C8 aqueous solution exhibits a similar tendency, however, with a much lower COF with a minimum of 0.02 (Figure 9b). As the applied pressure increases, the frequency at which the minimum of the COF is observed shifts to higher sliding velocities.

## 4. Discussion

Friction tests in which alkyl ß-D-glucopyranosides C6–C10 are added to water exhibit strongly reduced friction and, accordingly, much less wear. The extent to which friction is reduced, however, depends quite strongly on the conditions. One important parameter is the additive concentration.

At very low concentrations, the surfactant molecules become adsorbed to the steel surface to form a boundary lubricant layer. At the CMC, the individual molecules begin to aggregate to form micelles in solution, driven by hydrophobic force, so that the contact between the hydrophobic tails of the surfactant and water is limited [51,52]. The CMC decreases with the increasing chain length of the alkyl group, from 250 mM for C6 to 25 mM of C8 and 2.2 mM for C10. The adsorbed layer of surfactant molecules at the interface reduces the friction effectively compared to pure water. The resulted COF remains constant at COF = 0.14 until the concentration reaches 20%, where a strong decrease in the COF is observed (Figure 5). The friction regime can be described by the lambda ratio ($\lambda$), which represents the ratio of the minimum film thickness ($h_{min}$) to the composite surface roughness ($R_{a1}$ and $R_{a2}$) [53]

$$\lambda = \frac{h_{min}}{\sqrt{R_{a1}^2 + R_{a2}^2}} \tag{1}$$

The film thickness is estimated according to a simplified form of the Reynolds equation [54]

$$h_{min} = \sqrt{\frac{C_1 \cdot L \cdot \eta \cdot v \cdot B^2}{F_n}} \tag{2}$$

where $C_1$ is a prefactor determined by the sliding object's inclination ($C_1 = 2$), $\eta$ the dynamic viscosity, $v$ the sliding speed, $F_n$ the exerted normal force ($F_n = 50$ N), and B and L the characteristic values for the surface area. During the measurement, the friction partner oscillates periodically and sinusoidally on the steel surface, which results in a cyclic change of sliding speed from 0 to 0.16 m/s with a mean value of 0.1 m/s. Taking 1.4 and 29.7 mPa·s as the viscosity for the 10% and 40% C8 solution [50] and assuming the surface roughness $R_{a1} = R_{a2}$ after the friction test, which equals 0.48 and 0.48 μm lubricated by 10% and 40% solution, the lambda ratio is calculated to be 0.3–41 and 1.5–189 at low and high concentrations, respectively. At a low concentration, the lubricant layer acts as a boundary lubricant, where the steel surface asperities are occasionally in contact, as the minimum film thickness is lower than the average film thickness or the average surface asperities. At higher concentrations (above 20%), the higher viscosity induced a stronger hydrodynamic uplift effect to avoid asperities contact. The lubrication system shifts towards the "thin film lubrication" [50], thus it reaches a lower friction under high pressure, as the lubricant molecules become oriented in the friction gap. This view is supported by the value calculated for the lambda ratio, which is much higher than 1 in this regime.

The influence of alkyl chain length on the friction coefficient behavior is quite pronounced. With the shortest substituent, i.e., a hexyl group, it can be assumed that the molecular orientation in the friction gap is low. The increased friction gap with the sliding speed causes a steady decrease in COF (Figure 6). It is interesting to note that the COF of the C8 and C10 aqueous solutions is comparable despite the large difference in dynamic viscosity. Matsson et al. have reported that different alkyl glucopyranosides possess similar adsorption behavior on hydrophilic metal surfaces, independent of their alkyl chain length [55]. However, C10 molecules form a liquid crystalline phase with an ordered structure at a high concentration (Figure 4). It is supposed that this anisotropic property facilitates the molecule orientation in the shear direction. As a result, the viscosity in sliding direction is largely reduced. However, when the surface is lubricated by a C10 solution after washing all lubricant is removed and only a wear track is left (Figure 7c). Probably the layer is washed away by ethanol. Our measurement depicts a synergic effect of surface–surfactant interaction and molecule orientation under a high shear rate on friction. Previous work has confirmed that the friction of 40% C8 aqueous solution is low and its tribological mechanism can be described by the "thin film lubrication" model, whose features are a surface-bound lubricant layer and a fluid film with anisotropic viscosity [50,56–59]. The strong surfactant–surface interaction ensures the formation of a stable lubricating film. This demonstrates that an anisotropic viscosity can be achieved under a high shear rate due to the strong orientation of C8 molecules along shear direction and is conducive to low shear stress and friction [50].

The friction coefficient behavior of AGPs was compared to Glucopon 600 CSUP. Sulek et al. have studied the tribological property of Glucopon 600 CSUP and the effect of varying friction pairs [48,49]. They showed that it can effectively reduce motion resistance and wear, and a low friction below 0.1 is achieved after reaching the CMC (CMC = 0.05%). They proposed that the mixture of different (oligomeric) alkyl sugars has high surface activity, thus the high concentration of the surfactant in the surface phase results in the modification of the surface layer by the formation of mesophases, which leads to low friction and wear. In our measurement, the COF of Glucopon 600 CSUP reaches 0.06 under proper conditions. As it is composed of APGs with varying lengths of the alkyl chains a friction behavior between that of C6 and C8 solutions is observed.

The comparison of the frictional properties with Optigear 32 shows that the lubricant based on glucopyranoside has an even lower COF than that of such a standard lubricant oil. Thus C8 might be a candidate for a lubricant for a broad spectrum of technical applications, especially in the field of bearings with operation conditions at a low pressure and intermittent sliding speed.

Some problems are still to be solved, such as avoiding water evaporation, which changes the concentration of the lubricant. This is detrimental for such a system, since the friction coefficient behavior is dependent on the concentration. Thus efforts have to be made to keep the lubricant concentration more or less constant. This could be achieved by either having a fully encapsulated system or be adding water from time to time. Another problem to be solved is to avoid corrosion, which is inevitable for a steel surface exposed to a water environment. Here we are working on active galvanic protection

## 5. Conclusions

The friction behavior of lubricants based on glucopyranosides shows a strong dependence on its concentration and chemical structure. The sugar-based surfactant system was chosen for its biodegradable properties, as it might be environmentally benign. The concentration of the alkyl sugar in water should be high enough to ensure a sufficient lubricating film thickness, preventing asperities' contact. The length of the alkyl chain affects the dynamic viscosity, surface–surfactant interaction and the ability of molecule orientation under shear stress, which synergistically influence the friction performance. The study of the friction mechanism, indicates that the studied aqueous system has the potential to be an attractive additive in water as it reduces the friction by more than 95%. The tribological performance compares even very favorably with a commercial lubricant. As the lyotropic liquid crystalline solutions show promising tribological properties when the alkyl sugar is used as an additive, further investigations will be carried out in combination with a fully formulated, water-based lubricant. In particular, its interaction with polyalkylene glycol will be investigated tribologically. As the application case in these investigations, plain bearings will be used.

**Author Contributions:** T.A., A.K. and J.R. conceptualized the study and administered the project; W.C. and T.A. performed the experiments; W.C., T.A. and J.R. analyzed the data and wrote the manuscript. All authors have read and agreed to the published version of the manuscript.

**Funding:** This research was funded by MFW-BW (Ministerium für Wirtschaft, Arbeit und Wohnungsbau Baden-Württemberg, Project: BioSis).

**Acknowledgments:** The authors thank Susanne Beyer-Faiß (Co. Tillwich GmbH Werner Stehr) and Maria Ahrens (Fa. Iolitec GmbH) for many helpful discussions. This study was in part also supported by the Deutsche Forschungsgemeinschaft (DFG, German Research Foundation) under Germany's Excellence Strategy—EXC-2193/1–390951807 (livMatS).

**Conflicts of Interest:** The authors declare no conflict of interest.

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
