# Peer review of "Macroscopic Friction Studies of Alkylglucopyranosides as Additives for Water-Based Lubricants"

_lubricants, doi:10.3390/lubricants8010011_

Round 1
Reviewer 1 Report
This manuscript investigated the tribological performances of alkyl glucopyranoside as additives for water.
This manuscript fully explained the research background. The results were also interesting and suggested the applications under the low contact pressure (~1.2 MPa).
However, there are some questions to improve the quality of this investigation.
1) Wear volume
There is a photograph of the wear scar, but did you evaluate the amount of wear quantitatively?
2) Discussions
Is there any correlation with the evaluation of adsorption amount by using QCM etc?
3) References
If possible, add MDPI literature (such as Lubricants, Coatings, and Materials) to "References"
Reviewer 2 Report
The authors should address the following questions and comments:
Which are the specific applications for the lubricants proposed? Argue the requirements of those applications. How were the aqueous solutions prepared? A detailed description is required to reproducibility in other labs. Why was the ring-on-plate tribological test selected for the tribological evaluation? It is not representative for sliding bearings. How were the tribological test conditions selected and stated? How was the lubricant sample applied in the test? A test set-up schematic view should be provided to understand the lubrication technique employed. Figure 2 is not clear. I suggest indicating each curve with arrows or different vignettes for each lubricant sample results. It is not usual that mineral or synthetic oils exhibit non-newtonian behaviour. You are reporting that Optigear 32 oil presented a non-Newtonian behaviour. Is it typical for this lubricant?. Did you perform repetition rheological tests to confirm the behaviour discussed for all the samples? Wear rates are missing. Wear mechanisms are not enough to discuss wear behaviour. I suggest discussing what is the influence of having a non-Newtonian rheological behaviour (the case of most of aqueous solutions made) on COF and wear. Equations 1 and 2 comprises dynamic viscosities from Newtonian fluids, but the lubricants tested did not exhibit this behaviour. Why did you chose 1.4 mPa·s and 29.7 mPa·s as the viscosity for the 10 % and 40 % C8 solutions?. I think that the lambda model is not applicable for these lubricants.Author Response
see attachment

Reviewer 3 Report
The authors investigated the lubrication performance of water-based lubricants containing glucopyranosides. The effects of concentration and alkyl chain length have been studied. The work here is interesting and certainly, will draw a lot of interest from researchers working on novel water-based lubricants. There are few questions need to be addressed:
In section 3.2, a low friction coefficient was achieved when the C8 concentration was higher than 20%. It can be observed that there is a ~250s of the running-in period before the low friction. Does it correlate with the tribofilm or adsorbed layer formation?
In Figure 6, can the author provide the images of the wear on counter surfaces? As well did the author conduct any chemical characterization on the wear scar to analyze the composition of tribofilms?
Round 2
Reviewer 2 Report
Tribological investigations involve both friction and wear studies. In this reviewed paper version, the authors do not add wear rates or volumes results and discussions. So, this paper is aimed to report and discuss only on friction coefficient and poor evidence of wear. The title should be changed to "Investigation of Improved friction coefficient of alkylglucopyranosides as additives for water"
Change the caption of figure 5. Replace "tribological behavior" by "Friction coefficient behavior".
All the discussion sentences and figure captions refering to "tribological behavior" should be reviewed and changed to "Friction coefficient behavior" since the are supported only by COF results.
